# Using Neck Muscle Afferentation to Control an Ongoing Limb Movement? Individual Differences in the Influence of Brief Neck Vibration

**DOI:** 10.3390/brainsci13101407

**Published:** 2023-10-01

**Authors:** Maria Alekhina, Goran Perkic, Gerome Aleandro Manson, Jean Blouin, Luc Tremblay

**Affiliations:** 1Faculty of Kinesiology & Physical Education, University of Toronto, 55 Harbord Street, Toronto, ON M5S 2W6, Canadagoran.perkic@mail.utoronto.ca (G.P.); 2School of Kinesiology and Health Studies, Queens University, 28 Division Street, Kingston, ON K7L 3N6, Canada; 3Centre National de Recherche Scientifique and Aix-Marseille University, 3 Place Victor-Hugo, 13331 Marseille CEDEX 3, France; jean.blouin@univ-amu.fr

**Keywords:** neck, proprioception, vibration, pointing, individual differences, online control

## Abstract

When preparing and executing goal-directed actions, neck proprioceptive information is critical to determining the relative positions of the body and target in space. While the contribution of neck proprioception for upper-limb movements has been previously investigated, we could not find evidence discerning its impact on the planning vs. online control of upper-limb trajectories. To investigate these distinct sensorimotor processes, participants performed discrete reaches towards a virtual target. On some trials, neck vibration was randomly applied before and/or during the movement, or not at all. The main dependent variable was the medio-lateral/directional bias of the reaching finger. The neck vibration conditions induced early leftward trajectory biases in some participants and late rightward trajectory biases in others. These different patterns of trajectory biases were explained by individual differences in the use of body-centered and head-centered frames of reference. Importantly, the current study provides direct evidence that sensory cues from the neck muscles contribute to the online control of goal-directed arm movements, likely accompanied by significant individual differences.

## 1. Introduction

Neck muscle afferentation has been shown to be essential for fundamental sensorimotor control processes, including orientation, balance, and limb coordination (e.g., [1,2,3,4]). Anatomically speaking, the importance of the neck muscles for motor control likely stems from the high density of mechanoreceptors and muscle spindles located in the muscles and joints of the neck (e.g., [5,6]). Accordingly, it has been argued that the head serves as an ultimate and/or an intermediate reference for the location of targets in space and preparation of goal-directed movements (e.g., [7,8,9]). Specifically, changes in the sensed head-on-body position during upper-limb movements affect the perception of a target location (e.g., [10]) and the positions of body segments relative to each other (e.g., [2,11]). Similarly, neck vibration and changes in head position have been shown to alter the perception of “straight ahead” (e.g., [12,13]), which can also alter the perceived location of the target and the direction of pointing movements (e.g., [11]). Although neck proprioceptive input has been shown to contribute to the planning of movements during relative head/trunk rotations (e.g., [14]), no studies have investigated the contribution of cervical muscle signals for the planning vs. online control of discrete, goal-directed, upper-limb movements. In this light, the overarching goal of the present study was to provide new insights into the relative contribution of cervical afferents regarding these fundamental mechanisms of voluntary upper-limb movement control.

### 1.1. Neck Muscle Vibration Affects Pointing

Researchers have provided convincing evidence that proprioceptive information arising from the neck muscles contributes to visual space representation and reaching movements [10]. In their experiment, the left posterior neck muscles were vibrated, while participants repeatedly reached towards a visual target presented in a dark environment. The vibration of the left posterior neck muscles produced a false indication that these muscles had lengthened, resulting in the perception of a right-head rotation. The authors found that in 9 of the 10 participants, vibration induced a perceived motion of the visual target away from the side of vibration (i.e., to the right). Importantly, pointing movements towards the target were also biased by the neck vibration as the mean angular displacement of movement endpoints was shifted to the right of the body midline by 2.7° to 8.4°. While the study reported that the observed visual illusions were not related to movements of the eyes or head, the authors hypothesized that the illusion of the visual displacement of the target resulted from changes in perceived gaze direction via central mechanisms [10]. Because the participants’ eyes remained fixed on an earth-fixed target, the signal of head rotation induced by the neck vibration could have resulted in a sensed motion of the gazed target. To test this hypothesis, they evaluated subjective straight ahead (SSA), which should be in line with perceived gaze direction. They asked subjects to identify when a visual stimulus moving from 15° of eccentricity and towards the body midline was aligned with their SSA. The authors found that participants’ SSA was usually shifted to the left of the objective body midline during neck muscle vibration. The authors thus proposed that altered proprioception from the neck muscles primarily influences visual orientation and motion perception while leaving unaltered the perception of head posture relative to the torso.

Similar to the above study, others have reported that posterior neck vibration induced illusory target displacements without conscious perception of head motion [13]. However, in that study, participants also estimated the position of their nose with their unseen finger. It was found that participants made errors in the same direction as they perceived the target to be displaced and that the magnitude of the head and target motion illusions were correlated (R = 0.76). However, the between-participant variability was larger for the head than for the motion perception. Taken together, the results of the aforesaid studies suggest that individual differences in head and target motion as a result of neck vibration could influence reaching performance. Nevertheless, both studies showed the importance of neck afferent signals in encoding head and target positions (While it has been reported that perceived target motion effects disappeared or became less effective in a structured visual context [10], there is evidence that even in enriched visual contexts, neck muscle vibration may influence reaching movements [15]).

In the current study, we investigated individual differences in both the dominance of sensory modality processing as well as the specific muscles being primarily targeted. On one hand, it has been shown that neck vibration can lead to the interpretation of either a target or a body motion [16]. Specifically, these authors speculated that individuals may have predominantly processed visual or proprioceptive information, when experiencing illusory target motion or illusory head/body motion, respectively. Alternatively, considering the size of the vibration devices relative to the size of the targeted muscles, as well as their anatomical arrangement, it is possible that different muscle group combinations were stimulated in that study [16]. Indeed, the directional nature of limb endpoint displacements and perceived illusory motions are likely to be dependent on the specific group of neck muscles selected for stimulation. Anteroposterior endpoint pointing alterations and illusory target motion, in either an upward or downward direction with respect to the target, have been reported with bilateral vibrational stimulation of the sternocleidomastoidei (SCM) or of both the splenii capitis (SPL) and trapezius, respectively [17,18]. Similarly, mediolateral endpoint upper-limb reaching alterations, with either a rightward or leftward bias, have been reported when the right SCM and left SPL or the left SCM and right SPL were vibrated, respectively [17]. These effects are in accordance with the perceptual illusion of muscle lengthening that accompanies muscle vibration [19]. In sum, regardless of the preferred explanation, individual differences in sensory processing were expected in the current study and thus considered from the outset.

Overall, to the best of our knowledge, studies that have tested the effect of neck vibration on goal-directed arm movements did employ prolonged vibration, which is required to elicit perceptual biases (i.e., perception of head and/or target motion). These studies unambiguously showed that somatosensory cues arising from the neck muscles can influence the planning of goal-directed movements. However, it is not known if and how these cues related to the body–target relationship can influence arm movements once they have been initiated (i.e., influence the online control of voluntary actions)—that is, within the relatively shorter timeline of a discrete upper-limb reaching movement (i.e., less than 1 s).

### 1.2. Online Control of Voluntary Actions

When reaching for a visual target, the retina-coded target location must be combined with the eye and head position to derive its spatial location with respect to the body and program the upcoming motor commands (e.g., [20,21]). Disruptions in visual signals after movement initiation can significantly alter limb trajectories (see [22,23,24,25,26]). Other sensory modalities also appear to contribute to ongoing goal-directed actions (e.g., [27]). Researchers have tested the involvement of the vestibular inputs by stimulating the labyrinths through galvanic vestibular stimulation (GVS) at the onset of goal-directed movements [28]. When exposed to GVS, participants performed rapid adjustments in the movement direction (see also [29,30,31]). Adjustments to vestibular perturbations were often attributed to the perception of head motion [32]. Similar to vestibular signals, and as noted above, neck proprioception could also be used in the detection of head and gaze motion and body position localization, to ultimately influence the online control of upper-limb movements.

In the current study, we took advantage of the well-known effect of neck vibration on the internal representation of the body configuration to test whether neck somatosensory cues can be used online (i.e., during movement execution) to update the relative positions between the body and the target, which would in turn alter the motor commands of the upper limb. To achieve this goal, we employed brief neck vibrations (i.e., <2 s) to induce perceptual biases in head–trunk relative positions. Two vibration periods were employed: one period covered the movement preparation stage (Planning) and another period covered the movement execution stage (Online). Using an orthogonal manipulation of both neck vibration periods, we employed four experimental conditions (i.e., None, Planning, Online, and Complete: see below for details). Similar to previous studies (e.g., [14]), we hypothesized that neck proprioception would contribute to the planning of goal-directed actions and result in early trajectory biases, but these trajectory biases would be corrected by movement end, via online sensorimotor processes. Likewise, hypothesizing that neck somatosensory cues can be used to amend an ongoing movement, neck vibration during movement was expected to result in significant trajectory biases in the latter portions of the movement. Finally, we anticipated the largest trajectory biases when neck vibration was presented during both the planning and online control of a goal-directed movement.

## 2. Methodology

### 2.1. Participants

The experiment involved 19 participants (14 males and 5 females, aged 21–25 years). They were all right-handed, right-eye-dominant, and reported normal or corrected-to-normal vision. All participants also had to self-declare being free of neck and upper-limb injury. The study was approved by the University of Toronto Research Ethics Board and all procedures were conducted in accordance with the 1964 Declaration of Helsinki. All participants gave informed consent before inclusion in the study and they were compensated CAD 10 for their time.

### 2.2. Apparatus

During the experiment, participants were seated upright on a chair in total darkness in front of a table surface and a half-silvered mirror (see Figure 1). On the table surface, a small piece of rough cloth (1 by 1 cm) indicated the starting position for the right index finger, which was located 13 cm to the right of the participant’s midline. When looking straight ahead, participants saw the reflection of a green light-emitting diode (LED: 2 mm in diameter) that served as the target aligned with their body midline. The virtual location of the LED was at a distance of ~50 cm from the eyes, 12 cm below eye level. The half-silvered mirror allowed participants to see the lit LED target, while preventing the vision of their aiming limb throughout the movement (i.e., to avoid feedback, adaptation, and learning; see [33]). In addition, as in previous studies (e.g., [10]), participants viewed the target monocularly through a vertical slit in an opaque panel in front of the right eye. The slit subtended 1° of the visual angle and was aligned with the visual target, such that the target would disappear from the participant’s view if the head or eye had moved. Notably, no participants noted losing sight of the target. In addition, this monocular viewing method also had the advantage of avoiding the use of a chin rest that would alter the normal head support and associated neck muscle activity.

An infrared-emitting diode (IRED) was attached to the tip of participant’s right index finger and was sampled at 500 Hz by an Optotrak Certus system (Northern Digital Inc., Waterloo, ON, Canada) during each trial (2 s in length). Neck vibration was provided by two customized high-intensity cylinder-like vibrators (38 mm diameter, 75 mm long, 125 g, excursion 0.5 mm, 80 Hz: Model VB100, Dynatronics, Valence, France). The vibrators were attached to the participant’s neck using athletic pre-wrap tape (Red Lion Prewrap). A 1.5 cm diameter cylinder was mounted on each vibrator to selectively stimulate the targeted muscles specified below. Vibration onset and offset was controlled using an analog-to-digital board (PCI-6024E, National Instruments Co., Austin, TX, USA) and a custom-made controller that allowed us to reach a 80 Hz vibration and to stop the vibration in less than 100 ms. A custom MATLAB program gathered the Optotrak IRED displacement data and controlled the target LED and vibrators in real time.

The right splenius capitis (SPL) and left sternocleidomastoid (SCM) muscles were chosen for simultaneous vibration [17]. For the SPL muscle, the position of vibration was on its belly, located at approximately 3.5 cm from the spine. For the SCM muscle, the position of vibration was also on its belly, at approximately 40% of its length below its insertion at the mastoid (i.e., ~5 cm from the mastoid apex). Previous research has shown that the concurrent vibration of the right SPL and left SCM produces the illusion of a turning head motion to the left [17,34,35,36].

### 2.3. Experimental Procedure

Participants were first familiarized with the task of pointing to the target in darkness without neck vibration (20 trials). Participants were instructed to reach to the virtual target location in space as accurately and as fast as possible, within 1 s. Before the beginning of each trial, participants positioned their right index finger on the home position and waited for a 1s target preview, whose onset was accompanied by an auditory signal (i.e., three 30 ms beeps separated by 30 ms). After the 1 s preview offset, the data collection started; then, the target reappeared 400 ms after, and, after another 850 ms, a single auditory beep (i.e., 50 ms beep) signaled to participants to reach to the target position. After movement end, the participant was instructed to keep their index finger still until a “return” auditory signal (two 50 ms beep separated by 50 ms) was heard, which coincided with target offset and trial end.

After the familiarization, participants performed 20 consecutive trials without vibration (i.e., Baseline trials). After the Baseline trials, the main experimental phase involved 80 trials collected using a pseudo-random order, ensuring that each of the 4 experimental conditions (i.e., None, Planning, Online, and Complete) were presented 20 times each and no more than 2 times in a row. In this experimental phase, participants performed reaching movements following the same general procedure with the neck vibration applied either (i) before movement initiation (Planning condition), (ii) during movement execution (Online condition), (iii) throughout both phases (Complete condition), or (iv) not at all (None) (see Figure 2). In terms of timing in the Planning condition, vibration started at target onset and lasted until movement start. In the Online condition, vibration started at movement start and lasted until movement end. In the Complete condition, vibration started at target onset and lasted until movement end. The movement start and end were detected using real-time Optotrak data. On average, the vibration lasted ~800 ms during both the Planning and Online conditions, and ~1600 ms during the Complete condition. In the None condition, there was no vibration and the timing was identical as for trials performed in the Baseline conditions (see Figure 2). Finally, after the experimental phase, participants performed another block of 20 trials without vibration (i.e., Final trials), which was similar to the Baseline trials.

### 2.4. Data Reduction and Analyses

First, to examine if exposure to vibration led to carry-over effects on the spatial aspects of the limb trajectory, we contrasted the mean spatial finger position in the directional (i.e., medio-lateral) axis at each 20% of the MT for the None condition vs. the Baseline and Final trials. The absence of significant differences between no-vibration trials (i.e., None, Baseline, and Final) across movement proportions—including movement offset—would confirm that the use of None condition trials was an adequate control condition for the experimental trials.

More importantly, based on the current literature, we anticipated significant directional (i.e., medio-lateral) upper-limb trajectory differences across the neck vibration conditions (e.g., [10,36]). As such, our main dependent variable was the directional bias (DB, i.e., in the medio-lateral axis) of the reaching trajectories performed in the experimental conditions of interest (i.e., Planning, Online, Complete), as compared to the trajectories observed in the control experimental condition (i.e., None).

The DB variable was calculated for each participant individually using the difference in mean spatial finger position throughout the movement between each of the Planning, Online, and Complete conditions and the corresponding mean spatial finger position in the None condition. Specifically, the difference in the average DB (i.e., $DB\;\bar{x}$) of each experimental condition vs. no-vibration trials was computed at each 5% of movement time (MT) to detect any individual differences (see below). Afterwards, the $DB\;\bar{x}$ at each 20% of MT were used for the statistical analyses.

The other dependent variables that were computed to ascertain any speed–accuracy trade-offs (see [37]) and basic movement planning vs. online control mechanisms (see [22]) included movement time (MT), time to peak velocity (i.e., acceleration phase duration, TAccel), time after peak velocity (i.e., deceleration phase duration, TDecel), and peak velocity (PV). The MT was quantified as the time between movement start and end, based on when the instantaneous resultant limb velocity (i.e., across all movement axes) was above and below 30 mm/s and stayed below this level for 3 and 5 consecutive samples (i.e., 6 and 10 ms), respectively. PV, TAccel, and TDecel were also calculated from the resultant velocity profile. PV was estimated as the maximum velocity achieved during the movement (i.e., between movement onset and offset). TAccel was the time between movement onset and maximum velocity, and TDecel was defined as the time between maximum resultant velocity and movement offset.

Because individual differences in the reaching responses were deemed possible and likely (e.g., [16]), participants were parsed based on the limb trajectory biases (DB). We computed the area under each trajectory bias curve for each participant as an arithmetic sum of the signed biases at each 5% of the MT, yielding one summed trajectory bias value ($\sum DB\;\bar{x}$) for each of the vibration conditions (see Figure 3 for sample trajectory bias graphs). These signed sums of biases were then added, yielding a single composite trajectory bias score (CS) for each participant. Through visual exploration of the data, CS was deemed as the best proxy to assess typical participants’ reaching movements because it mirrored the Planning condition results and represented an intermediate participant segregation compared to the Online and Complete conditions (see Table 1). Negative composite trajectory bias scores represented leftward biases, while positive scores represented rightward biases. Thus, the participants were ultimately assigned to a Leftward or a Rightward group, according to the sign of their CS (see the Results section for the number of participants assigned to each group).

After the group assignment, the effect of the vibration conditions on the absolute mean directional trajectory biases ($\left|DB\;\bar{x}\right|$ calculated from the None condition) was analyzed at each 20% of MT. Specifically, the $\left|DB\;\bar{x}\right|$ data were submitted to a 3 (condition: Planning, Online, Complete) × 6 (movement proportion: 0%, 20%, 40%, 60%, 80%, 100% of MT) × 2 (group: Leftward, Rightward) mixed ANOVA. Moreover, each kinematic and temporal measure was submitted to a 4 (condition: None, Planning, Online, Complete) × 2 (group: Leftward, Rightward) mixed ANOVA.

All analyses employed mixed repeated-measures analyses of variance (ANOVA). Where appropriate, F-statistics were corrected for violations of the sphericity assumption using the Huynh–Feldt procedure, with the degrees of freedom reported to the nearest decimal. Tukey’s HSD post-hoc procedure was used to decompose all significant effects/ interactions involving more than two means. An alpha level of 0.05 was used to interpret all tests. All dependent variables’ data were reported using the mean and between-subjects standard error of the mean.

## 3. Results

The individual differences analysis indicated that the neck muscle vibration typically induced leftward trajectory bias in 12 participants (~63% of the participants) and typically exhibited rightward trajectory bias in 7 participants (~37% of participants). Accordingly, the 12 participants who exhibited leftward trajectory bias formed the Leftward group, and the 7 participants who exhibited rightward trajectory bias formed the Rightward group (see Table 1). This assignment was then employed to contrast the absolute composite score ($\left|DB\;\bar{x}\right|$) from the limb trajectory data across these groups.

The analysis of $\left|DB\;\bar{x}\right|$ yielded main effects for the condition (F_2, 34_ = 3.89, *p* < 0.03) and movement proportion (F_3.9, 60.8_ = 11.86, *p* < 0.001), as well as the movement proportion by group (F_3.6, 60.8_ = 3.81, *p* < 0.01) and movement proportion by condition by group interactions (F_7.3, 123.6_ = 2.18, *p* < 0.04). For brevity purposes, we opt to only report on the highest-order interaction. Post-hoc analyses (HSD = 5.3 mm) of this three-way interaction (see Figure 4) yielded differences both within and between conditions in each group, as well as between the groups. Comparisons within each condition in each group revealed that the Leftward group exhibited greater $\left|DB\;\bar{x}\right|$ with respect to movement start: (i) in the Planning condition, at 40% of MT (7.1 ± 1.2 mm); (ii) in the Online condition, at 40% of MT (5.8 ± 1.4 mm); and iii) in the Complete condition, at 40%, 60%, and 80% of MT (12.0 ± 2.3 mm; 8.8 ± 1.6 mm; and 6.6 ± 1.5 mm, respectively). In contrast, the Rightward group exhibited greater directional bias with respect to movement start in the Complete condition at 100% of MT only (7.7 ± 1.6 mm). Comparisons made between conditions in each group revealed that $\left|DB\;\bar{x}\right|$ was greater in the Complete condition than in the Online condition at 40% of MT for the Leftward group only. Last, the post-hoc analyses yielded a significant difference between the groups in the Complete condition only at 40% of MT: the Leftward group ($\left|DB\;\bar{x}\right|$ = 12.0 ± 2.3 mm) was more biased than the Rightward group ($\left|DB\;\bar{x}\right|$ = 2.6 ± 3.0 mm).

The analysis of MT yielded no main effects (Fs < 1.2, *p*s > 0.3) but a significant condition by group interaction (F_3, 51_ = 4.45, *p* < 0.008). The post-hoc analysis (HSD = 32 ms) showed that the Leftward group exhibited longer MT than the Rightward group in the Online (Leftward = 795 ± 20 ms; Rightward = 737 ± 27 ms) and Complete conditions (Leftward = 795 ± 22 ms; Rightward = 742 ± 29 ms) but not in the Planning condition (Leftward = 775 ± 21 ms; Rightward = 745 ± 28 ms) or the None condition (Leftward = 774 ± 23 ms; Rightward = 766 ± 27 ms).

The analysis of the TAccel revealed only a main effect for condition: F_3, 51_ = 3.07, *p* < 0.03. Specifically, participants in both groups spent more time reaching peak velocity in the Online condition (TAccel = 225 ± 11 ms) compared to the Complete condition (TAccel = 211 ± 10 ms), but not relative to the None and Planning conditions (TAccel = 221 ± 11 ms; 216 ± 10 ms, respectively) (HSD = 11 ms). No other main effects/interactions were found (Fs < 1.4, *p*s > 0.3). The analysis of TDecel yielded no main effects (Fs < 2.5, *p*s > 0.3) but a significant condition by group interaction emerged: F_3, 51_ = 3.01, *p* < 0.04, HSD = 35 ms. Participants in the Leftward group spent more time between peak velocity and movement end than the Rightward group during the trials that included neck vibration (i.e., Planning, Online, Complete), as compared to the trials without vibration (i.e., None). The analysis of PV yielded a main effect for condition (F_2.4, 40.9_ = 10.89, *p* < 0.001). Participants in both groups performed their movements with lower PVs when vibration was provided during the movement (Online: PV = 1.51 ± 0.13 m/s) or not at all (None: PV = 1.51 ± 0.13 m/s) than in movements performed in the Planning (PV = 1.58 ± 0.13 m/s) and Complete conditions (PV = 1.58 ± 0.13 m/s) (HSD = 0.09 mm/s). No other main effects/interactions were found (Fs < 1.1, *p*s > 0.4).

## 4. Discussion

The present study was designed to determine if somatosensory information from the neck can be used to modify ongoing goal-directed arm movements. While individual differences were observed (i.e., early leftward or late rightward limb trajectory biases), they did not challenge our core hypothesis. As such, it was deemed appropriate to provide an explanation for these individual differences and how they relate to the use of cervical information for the planning vs. online control of a goal-directed movement.

### 4.1. Neck Vibration and the Online Control of Reaching

Our core hypothesis was that neck somatosensory information can contribute to the online control phase of a reaching movement. We also anticipated that neck vibration during the planning phase would induce trajectory biases that could be corrected prior to movement end. As a result, it was also hypothesized that neck vibration during the planning and online control phases (i.e., Complete) would exhibit the largest trajectory biases. These predictions allowed us to test for the capacity of the brain to process cervical cues for the online control of discrete, upper-limb reaching movements. Our hypotheses found some support, although, in addition to unexpected individual differences, the patterns of results were more complicated than anticipated.

First, the Leftward group exhibited significant trajectory biases at 40% of the trajectory in all neck vibration conditions. Second, the Rightward group exhibited significant trajectory biases at movement end in the Complete neck vibration condition only. More importantly, regardless of leftward or rightward directional biases or when such biases were observed during reaching movements, the influence of neck vibration on limb trajectories was significant if presented during both the planning and online control phases of the movement (i.e., Complete condition). These results provide support for the idea of the online control of arm movements using somatosensory information from the neck muscles. The individual differences in the directional biases and the associated explanations for these differences can lend further support to this core hypothesis.

### 4.2. Using Limb Trajectories to Segregate Participants

Individual trajectory biases in the vibration conditions showed that seven participants exhibited rightward movement deviation from the no-vibration trials, whereas the remaining 12 participants deviated to the left. It was shown that vibration of the left dorsal neck muscles, as used in the present study, produced pointing deviation contralateral to the direction of the visual illusion (i.e., to the left) in 10% of participants, while, in 90% of participants, pointing deviation was in the same direction as the visual illusion (i.e., to the right: [10]). One possible explanation as to why individual differences were more prevalent in the current study as compared to others (e.g., [10]) is that the neck vibration durations were much shorter (i.e., 0.8–1.6 s) compared to other neck vibration reports (i.e., at least 8 s; see [36]). Considering the different time spans of the sensorimotor and perceptual effects of muscle vibration (see [19]) and more particularly neck vibration (e.g., [38]), future research should aim to test if brief tendon vibration reliably assesses sensorimotor functions vs. cognitive functions (i.e., perception). Such research could provide further evidence that brief neck vibration aptly probes sensimotor functions and perhaps replicate individual differences in the observed patterns of limb trajectory biases.

### 4.3. Individual Differences in Prioritizing Sensory Cues

The contribution of multisensory cues (e.g., the relative contribution of visual and non-visual cues (see [20,39]): see also [16]) can potentially differ between individuals in both motor and cognitive tasks. It is possible that such sensory prioritization can explain the individual differences in the limb trajectory biases observed in the current study, which are supported by at least two lines of research.

Some of the empirical evidence for individual differences in sensory cue prioritization can be drawn from studies in microgravity. For example, cosmonauts could estimate the relationship between the body and the surrounding environment using either visual or bodily internal information [40,41]. Some cosmonauts were identified as being more “visuospatial” as they appeared to use the visual environment to establish their own spatial orientation and experienced significant discomfort when being in an “upside down” position relative to the visual environment. In contrast, other cosmonauts were identified as being more “internal” because they seemed to develop a representation of space based on their own internal body coordinates. Finally, other cosmonauts were classified as mixed. Altogether, when making perceptual and spatial orientation judgments, some individuals may rely more on visual cues while others may rely more on somatosensory and vestibular cues.

Another source of support for individual differences in the prioritization of sensory cues comes from experimentally induced attentional biases [42]. They asked individuals to perform the rod-and-frame test (RFT) under different instructional sets. In a series of trials, they instructed the participants to pay attention either to external visual cues or to internal body cues (e.g., “We have found that when you attack this problem [the RFT], it is quite important that you pay close attention to stimuli arising from inside (outside) your body”: see [42]). The results showed that participants who paid attention to an external environment performed in a more visual-field-dependent manner (i.e., their judgments were affected by the orientation of the frame). In contrast, when asked to pay attention to body cues, participants were less affected by the presence of a frame. Such attentional instructions were also shown to extend to the encoding of the head position during neck vibration [43], likely because illusory head motion was rarely perceived during neck vibration, unless the observer’s attention was drawn to the head prior to the vibration.

In sum, there is ample evidence of individual differences in prioritization in the encoding of visual and proprioceptive cues, including neck proprioception. It is also important to explore how different individuals may adapt across trials in the experimental conditions. However, critically, the observed individual differences could be leveraged to support the grouping approach employed in the limb trajectory analyses.

### 4.4. Different Patterns of Trajectory Biases and Sensory Prioritization

Participants in the Leftward group exhibited greater directional bias in the middle of the limb trajectories (i.e., 40% of MT) in all vibration conditions, while the Rightward group deviated the most at movement end (i.e., 100% of MT), only in the Complete condition. Notably, these statistical differences were obtained from the absolute deviation values and the average biases were in opposite directions across groups (see Table 1). From a mechanistic perspective, the inconsistencies in the direction and timing of movement deviations between the groups could be explained by different sensory integration processes by the central nervous system (CNS). Specifically, vibration of the left SCM/right SPL produced a false indication that these muscles had lengthened, indicating a leftward head rotation relative to the trunk (e.g., [34]). Because the eyes continued foveating the target, the signal about false leftward head rotation in all participants may have resulted in two types of misinterpretation of the multisensory input by the CNS, yielding corresponding limb trajectory biases (see also Figure 5). Nevertheless, it is important to mention that the sample size for the Leftward group was rather small (*n* = 7) and that these potential differences in sensory prioritization remain explorative at this time.

One type of misinterpretation can be associated with the prioritization of external or visual cues (Leftward group), which ironically facilitates visual illusions such as a target displacement. Thus, the performance biases in the Leftward group could be due to illusory target motion, resulting in biases earlier in the trajectory that could be corrected based on sensory information emerging from the body and limb later in the trajectory. One real-life explanation for a similar combination of a stationary retinal image of a target with neck movement would be observed during the tracking of a moving object using head rotation. Thus, for some participants, their CNS could have interpreted the target as moving with the head (i.e., for the Leftward group), which implied an interpretation that the target had moved to the left of the body. In such conditions, the fixated target appears to move in space because it would have to do so in order to remain motionless on the retina.

Another type of misinterpretation can be associated with prioritizing internal or somatosensory cues (Rightward group; see also [42,43,44]). Indeed, it is also possible that, for other participants, their CNS interpreted the same head shift to the left as the other group but that the target was encoded as remaining stationary in space. This alternate interpretation implies an illusory shift in the eye-in-head position to the right (i.e., Rightward group). For example, if there is a stationary target to fixate on, head movement normally causes reflex-compensatory eye movements (i.e., vestibular–ocular reflex). In such a scenario, the target would be encoded by the CNS to the right of the perceived head orientation, thus requiring rightward arm trajectory amendments. Such an explanation is corroborated by previous evidence that neck vibration can elicit target motion in the opposite direction to the anticipated perceived head rotation (e.g., [36]).

Additional support for differing CNS interpretations as a result of neck muscle vibration was reported in a study in which the target was presented either in a dark or structured visual environment [15]. In a dark environment, all participants reported target movement. However, in a structured visual environment, only three out of nine participants reported target movements, while another subset of three participants perceived slight head motion. The reduced reporting of target motion with a visually enriched environment could be explained by the CNS interpreting the motion of the head during neck muscle vibration instead of the target. Others have also reported that vibration either leads to a perceived target motion in the direction contralateral to the vibrated muscle site or a perceived ipsilateral rotation of the trunk [16]. The authors argued that the differing effects of neck muscle vibration can stem from a natural predisposition or a practiced utilization (e.g., with sports) favoring visual or kinesthetic processes. Nevertheless, it has previously been shown that neck vibration yields different perceptions across participants, which can also be influenced by the presence of surrounding visual cues.

The two different types of “sensory interpretations” that participants may have adopted in our experiment can also be used to explain the trajectory bias patterns observed in the vibration conditions. First, despite the different CNS interpretations of head orientation biases induced by neck vibration, our explanation maintains that all participants planned their movements with respect to the sensed target position relative to the body. Specifically, it is suggested that the initial impulse control is egocentric because it is based on an internal, body-based representation of the expected sensory consequences of the limb movement (e.g., [45]). This idea is in line with [22] impulse regulation mechanisms during goal-directed aiming movements. Specifically, the earliest online trajectory regulation processes relate to the prepared initial impulse and early sensory information, to regulate this impulse, if necessary. In this sense, impulse regulation processes may rely heavily on body-based representations. This early reliance on body-based references can explain why we observed greater leftward trajectory biases in all vibration conditions early in the movement for the Leftward group (i.e., because the target appeared for them to be to the left of the perceived body midline).

In contrast, the Rightward group did not exhibit significant trajectory biases to the right early in the movement, probably because the target appeared to be approximately in line with their body midline. However, as movements unfolded, all participants seemed to incorporate the head-centered representation of the target when controlling their ongoing movements. Specifically, it is hereby suggested that late movement control depends on the relative positions of the unseen limb and the target coded in a head-centered frame of reference. This idea is in line with proposed limb–target regulation mechanisms [22]. We purport that the CNS contrasts information about the limb and target (i.e., limb–target regulation) after impulse regulation during a goal-directed movement.

Based on the above mechanisms, it is also possible to explain why the Leftward group did not exhibit significant trajectory biases at movement end. Indeed, if limb–target regulation relies more on a head-centered frame of reference, then one can expect a reduction in trajectory biases from 40% to 100% of movement time. In contrast, for the Rightward group, if the target was perceived as it had moved rightward relative to the head, this would explain the late rightward limb trajectory corrections based again on this head-centered bias for the late limb–target regulation. Altogether, it is suggested that the early stages of a limb trajectory (i.e., impulse regulation processes) are more subject to the influence of a body-centered frame of reference, while the later stages (i.e., limb–target regulation processes) are more influenced by a head-centered frame of reference.

The suggestion that neck vibration influenced participants in different ways is also supported by the temporal parameter analyses. Specifically, the Leftward group exhibited longer movement times in the Online and Complete conditions than the Rightward group. This effect was mediated by the longer times spent after peak velocity (i.e., longer deceleration) in the Planning, Online, and Complete conditions by the Leftward group compared to their counterparts in the Rightward group. The fact that participants of the Leftward group spent more time decelerating in all vibration conditions is consistent with a discrete or pseudo-continuous feedback-based movement correction (e.g., [22,46,47]). A likely explanation for these corrections is that, for the Leftward group, the biased neck signals induced a sensed target displacement, which implies a leftward shift of the target in the body-centered frame of reference. In contrast, the Rightward group might have experienced a bias in the sensed head position, but no target displacement. As a result, the initial impulse and early online control phases of the movement were in the direction of the body midline, and head-centered target localization biases only induced small and late trajectory amendments, influencing the time spent after peak limb velocity to a lesser extent.

## 5. Conclusions

In summary, the present study provided novel evidence that neck muscle vibration as brief as 800 to 1600 ms induces reliable deviations in discrete, goal-directed arm movements. As such, measuring limb trajectory biases during neck vibration could be a useful proxy to assess the use of neck proprioception to control ongoing arm movements. Moreover, it is possible that the observed trajectory deviations were a result of complex multisensory processes associated with illusory head and/or target motion. Specifically, individual differences were observed in terms of trajectory biases, which could be explained by the reliance on body- vs. head-centered cues. In the majority of the tested individuals, the vibration of the right splenius capitis (SPL) and left sternocleidomastoid (SCM) muscles elicited early leftward changes in the hand trajectory, which we propose to result from a sensed shift in target location relative to the body. For the other subjects, neck vibration induced late rightward modification of the hand trajectory, which we purport could arise from a sensed shift in head position, influencing the sensed target location relative to the head. Together, the present results are consistent with the idea that the planning and the initial phase of arm movements rely to a greater extent on a body-centered frame of reference, while the later phase would rely more on a head-centered frame of reference. However, most importantly, and in line with the primary purpose of the current experiment, neck proprioception does contribute to the online control of goal-directed movements towards visual targets. Thus, whether considering high-performance sport applications (e.g., stick handling in a hockey game) or rehabilitation contexts (e.g., using a cane), neck proprioception does contribute to the control of voluntary actions.

## Figures and Tables

**Figure 1 brainsci-13-01407-f001:**
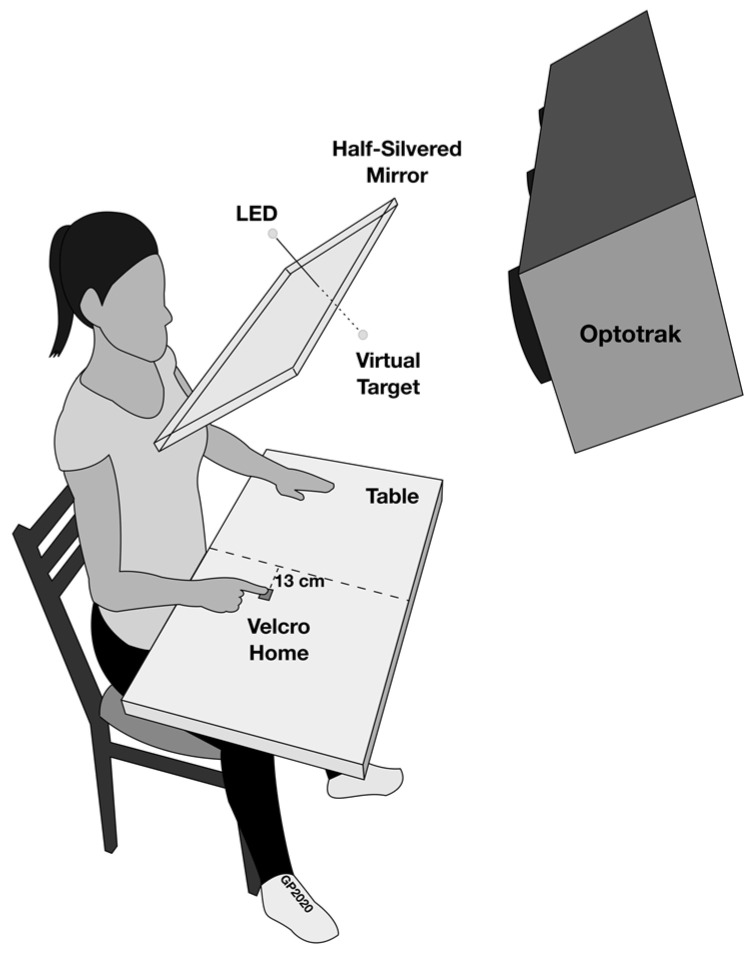
Depiction of the experimental apparatus (not to scale). Participants sat on a chair at a table. The dashed line on the table represents the participants’ midline, 13 cm from which was a starting position for their pointing finger (Home). In front of the participants was a half-silvered mirror, through which they saw a reflection of an LED (virtual target), which was aligned with the participants’ midline. Participants peered at the virtual target through a slit over their right eye (not shown), which subtended 1° of the visual angle.

**Figure 2 brainsci-13-01407-f002:**
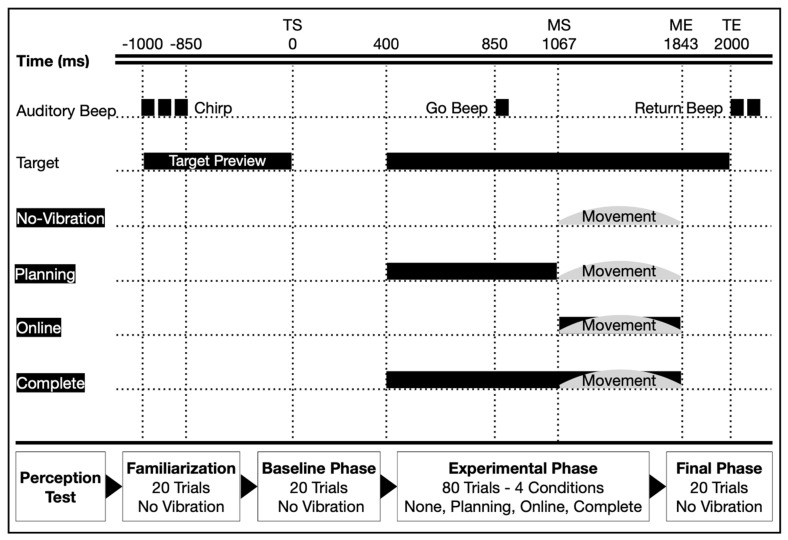
Depiction of a single trial in all four experimental conditions. At the top of the figure, the time scale reflects the chirp, target preview, trial start (TS) and end (TE), target/go-beep/return-beep, and movement start (MS) and end (ME). The figure contains the onset and offset of vibration in the no-vibration trials (i.e., Baseline, None, Final), as well as in the Planning, Online, and Complete vibration conditions represented by the black bars. The time values are grand averages across all participants and conditions. The bottom of the figure illustrates the experimental protocol and trial breakdown in its entirety.

**Figure 3 brainsci-13-01407-f003:**
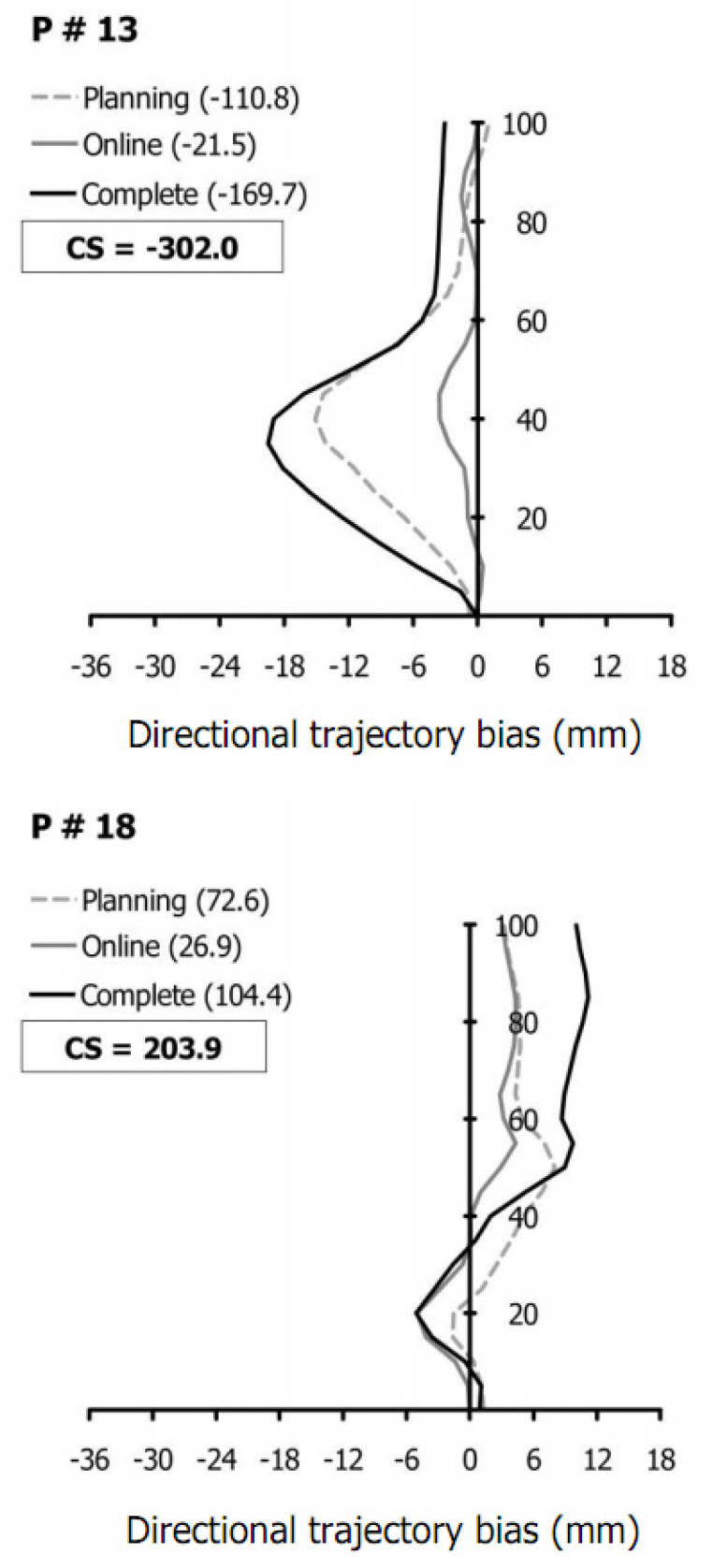
The individual directional trajectory biases ($DB\;\bar{x}$) as a function of the main conditions (i.e., Planning, Online, and Complete). Lines of graphs represent the participants’ average biases—for all trials—in the Planning (grey dashed), Online (grey solid), and Complete condition (black solid). Values in parentheses indicate $\sum DB\;\bar{x}$ for each condition. Negative CS represents individual composite score for the leftward bias (for one participant), whereas positive CS represents individual composite score for the rightward bias (for one participant).

**Figure 4 brainsci-13-01407-f004:**
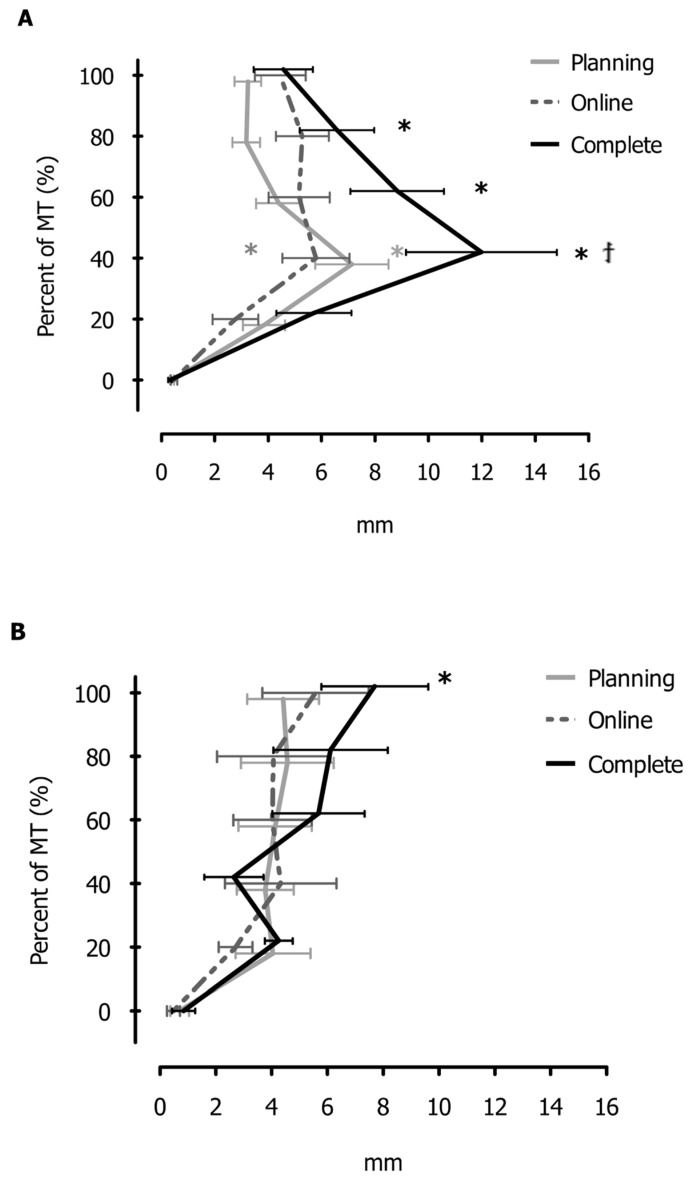
The absolute values of the mean directional trajectory biases ($\left|DB\;\bar{x}\right|$) at each 20% of MT as a function of the main conditions (i.e., Planning, Online, and Complete) for the Leftward group (**Panel A**) and Rightward group (**Panel B**), respectively. Error bars represent SEM. Asterisks (*) denote significant differences within each condition with respect to the movement start. Asterisks (†) denote significant differences between Complete and Online conditions at 40% of MT.

**Figure 5 brainsci-13-01407-f005:**
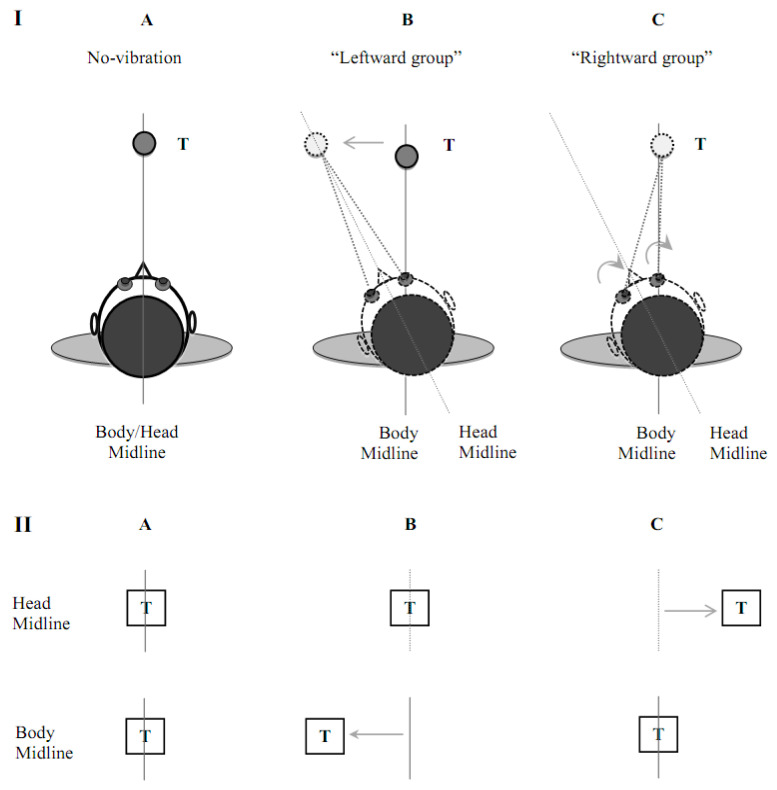
Illustration of two different sensory interpretations by the CNS in each group of participants (i.e., Leftward group and Rightward group). The top panels (**I**) depict bird’s eye views of the perceived head rotation and target location relative to the body midline. Panel **IA** depicts situation where all sensory inputs are congruent (no-vibration). Panel **IB** and **IC** depict multisensory inputs in the presence of vibration for the Leftward group and Rightward group, respectively. “T” indicates the target. The bottom panels (**II**) illustrate first-person perspectives of the perceived target position relative to the head and body midline in the no-vibration condition. Panels **IIA** depicts the situation where all sensory inputs are congruent (no-vibration). Panels **IIB** and **IIC** depict target position relative to the head and body midline in the presence of vibration for the Leftward group and Rightward group, respectively.

**Table 1 brainsci-13-01407-t001:** Summed directional trajectory biases ($\sum DB\;\bar{x}$) for the Planning, Online, and Complete conditions. CS represents the composite trajectory bias score for each participant. Negative CS represents leftward biases, while positive represents rightward. Participants are ordered from the largest leftward to the largest rightward CS.

Participant # /Condition	$\sum DB\;\bar{x}$ Planning	$\sum DB\;\bar{x}$ Online	$\sum DB\;\bar{x}$ Complete	CS	Group
P11	−86.6	−162.9	−377.5	−627.0	Leftward
P16	−125.3	−69.6	−188.7	−383.6	Leftward
P7	−98.1	−39.4	−223.0	−360.5	Leftward
P1	−114.6	−41.3	−151.2	−307.1	Leftward
P13	−110.8	−21.5	−169.7	−302.0	Leftward
P10	−5.4	−125.6	−96.2	−227.2	Leftward
P5	−16.3	−43.5	−103.1	−162.9	Leftward
P14	−69.1	−30.5	−61.4	−161.0	Leftward
P3	−28.4	−42.2	−75.1	−145.6	Leftward
P9	−115.7	89.8	−115.2	−141.1	Leftward
P8	−35.8	−19.8	−59.8	−115.4	Leftward
P2	−62.7	−14.1	8.3	−68.5	Leftward
P19	6.0	−9.8	17.9	14.0	Rightward
P4	26.9	−1.4	47.6	73.1	Rightward
P17	110.8	−5.1	9.5	115.3	Rightward
P15	124.2	1.8	68.1	194.0	Rightward
P18	72.6	26.9	104.4	203.9	Rightward
P6	51.6	138.7	70.2	260.4	Rightward
P12	78.7	178.5	199.5	456.7	Rightward

## Data Availability

Data cannot be shared due to the ethics protocol in place at the time of data collection.

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
