# Peer review of "Using Neck Muscle Afferentation to Control an Ongoing Limb Movement? Individual Differences in the Influence of Brief Neck Vibration"

_brainsci, 2023, doi:10.3390/brainsci13101407_

Round 1
Reviewer 1 Report
Review of the paper entitled : Using neck muscle afferentation to control an ongoing limb movement? Individual differences in the influence of brief neck vibration
The authors asked their participants to point at a target in the dark while vibrating their neck muscles, giving the illusion that their head and/or the target were rotating to the left. The timing of the vibrations - before and/or during the movement - was manipulated but had no clear effect. However, some subjects deviated to the right and others to the left in response to the vibrations. The authors suggest that this reveals differences in the use of proprioceptive and ocular information.
The article is clearly written and the ideas are interesting. I have several comments, some of which are major.
Major comments
The number of subjects is low (7 only in the rightward group). There is often a problem of reproducibility in the field and I would recommended adding a few more subjects to confirm the results and/or carrying out a power analysis to ensure that we can trust the results.
The authors analyzed the average of 20 trials. Is it possible that some subjects learned to compensate for the perturbations and not the others?
The timing of the deviations in the online and the planning condition did not differ. This suggests that the subjects perceived the target as being more on the left that it really was in both conditions when the movement started. Because the timing of the stimulation was not timed to the movement of the finger this suggests that the vibrations started before movement onset in the online condition too. The finger movement should be used to trigger (online) or stop (planning) the vibrations in order to establish a clear distinction between the conditions and to test the main hypothesis.
Minor comments
-The error bars are almost unreadable in Figure 4. Please also indicate whether the traces represent averages or single trials.
-How do you know that you did not vibrate the vestibular system ?
-L266 and elsewhere. What is a resultant velocity?
-L 322. You say that the “Leftward group exhibited greater |DB| with respect to movement start”. Do you mean with respect to the first 20% of movement time? I don’t understand actually how you computed the deviations at t = 0%, this should be zero (because the starting point is the same in all trials). Idem L 326. Movement end or 100% of MT… does it mean that 100% of MT is not the end? (or just chose one term maybe).
-L 445. You say that the “Rightward group deviated the most at movement end”. However, the amplitudes of these deviations were similar to that of the Lefward group at this time. Therefore, it is not clear that sensory processing was different in the two groups of subjects.
L 446-442. These information are interesting but maybe beside the point, I suggest to focus the discussion on the results of the present study.
The discussion is short compared to the introduction and should be improved.
Author Response
We thank Reviewer 1 for their positive and constructive comments on our manuscript. There were many important elements to clarify and refine. Below, we present the original comments as well as our responses. As well, please note that we have made a few more edits, including the citation, correcting a few typos, and removing two references not cited in the text. The updated manuscript format has been edited with track-changes, to highlight all changes since the last submitted version.
Reviewer 1
Review of the paper entitled : Using neck muscle afferentation to control an ongoing limb movement? Individual differences in the influence of brief neck vibration
The authors asked their participants to point at a target in the dark while vibrating their neck muscles, giving the illusion that their head and/or the target were rotating to the left. The timing of the vibrations - before and/or during the movement - was manipulated but had no clear effect. However, some subjects deviated to the right and others to the left in response to the vibrations. The authors suggest that this reveals differences in the use of proprioceptive and ocular information. The article is clearly written and the ideas are interesting. I have several comments, some of which are major.
Major comments
The number of subjects is low (7 only in the rightward group). There is often a problem of reproducibility in the field and I would recommended adding a few more subjects to confirm the results and/or carrying out a power analysis to ensure that we can trust the results.
It is worth mentioning though that the between group comparisons were not the focus of the study and that the groups were constructed based on individual differences in sensory processing that, we believe led to systematic differences in aiming performance. Thus, considering the pragmatic, technical, and ethical ramifications, we opted not to collect more data for the study. Specifically, based on our findings, it would be difficult to determine (using the same experimental setup) whether additional participants would be part of the rightward and leftward group prior to the experiment. Furthermore, for the between-group comparison of movement time (the analyses yielding a significant condition by group), we conducted a post-hoc power analyses using G*Power (see below for inputs) which found that the test was adequately powered (i.e., 99%). As post-hoc power analyses must be interpreted with caution, we have toned down the language and clarified the explorative nature of that portion of the analyses (see lines 472-474).
G*Power inputs: using estimated effect size of F = 1.3 , alpha = 0.05, sample size = 19, # of groups = 2 and repeated measures = 3 [Planning, None, Control], and a correlation of 0.8 among repeated measures.
The authors analyzed the average of 20 trials. Is it possible that some subjects learned to compensate for the perturbations and not the others?
This is yet another great question, so we performed another analysis to explore this empirical question. Specifically, we computed the spatial difference in the x-axis of the trajectory between the last and first trial of each condition. That allowed to determine if there was a compensation over the trials and detect any adaptation. Critically, that analysis revealed that one participant in each group had a potential adaptation effect in one condition. Indeed, one participant in each group did exhibit a larger difference in the x-axis than other participants between the first and last trial. However, these differences were likely not reflective of a compensation or adaptation. One participant in the rightward group exhibited an early rightward bias in the last vs. the first trial of the planning condition. And one participant in the leftward group exhibited a late larger bias in the online condition. As such, in two cases (i.e., one condition for one participant in each group) these changes did not reflect a compensation but rather a possible exacerbation of the effect. More importantly, considering that these 2 observations were made over 19 participants in 6 series of trials each (i.e., 2 occurrences out of 114 observations), we felt that such question and observations were falling short of statistical and theoretical relevance. However, we did flag the question of potential adaptation effects and individual differences (see lines 455-457).
The timing of the deviations in the online and the planning condition did not differ. This suggests that the subjects perceived the target as being more on the left that it really was in both conditions when the movement started. Because the timing of the stimulation was not timed to the movement of the finger this suggests that the vibrations started before movement onset in the online condition too. The finger movement should be used to trigger (online) or stop (planning) the vibrations in order to establish a clear distinction between the conditions and to test the main hypothesis.
It seems relevant to clarify our text because we did gather Optotrak data, analyzed it in real-time (see addition on line 194), and triggered the conditions based on the actual movement state. As a result, we also added a clarification that the timing of the conditions was based on real-time Optotrak data (see lines 226-227).
Minor comments
-The error bars are almost unreadable in Figure 4. Please also indicate whether the traces represent averages or single trials.
Thank you for this suggestion. We clarified in the figure caption that the traces were for the average of the trials (see line 296). And we hope that increasing the size of Figure 4 helped identify the error bars.
-How do you know that you did not vibrate the vestibular system ?
Very good question, which first led us to inquire about evidence of vestibular bias during vibrations. In patient populations, the skull vibration-induced nystagmus test does require direct vibration of the mastoid process of the temporal bone with a massage device (e.g., Dumas et al., 2014). Moreover, the effect of such vibration is a visual nystagmus. We vibrated the splenius capitis approximately 3.5 cm from the spine and the sternocleidomastoid ~5 cm from the mastoid apex. The vibrator weighed 125 g with a 0.5 mm amplitude at 80 Hz. These differences in the parameters give rise to significant doubt that our vibration units impacted vestibular activity. Be it more, if there were an impact of vibration, it would have been a nystagmus, which is inconsistent with the fact that all participants were able to maintain visual contact with the target through the slit (see lines 171-173). In sum, we have no compelling evidence or reasons to believe that the vestibular system was vibrated.
-L266 and elsewhere. What is a resultant velocity?
Thank you for asking. Resultant velocity means the velocity of the limb taking into account all movement axes (i.e., x, y, and z). This is now clarified on lines 269-270.
-L 322. You say that the “Leftward group exhibited greater |DB| with respect to movement start”. Do you mean with respect to the first 20% of movement time? I don’t understand actually how you computed the deviations at t = 0%, this should be zero (because the starting point is the same in all trials). Idem L 326. Movement end or 100% of MT… does it mean that 100% of MT is not the end? (or just chose one term maybe).
The DB values were calculated from the control/ no-vibration condition (see lines 251-255 & lines 256-260). As such, the deviations at 0% of MT were not necessarily an absolute zero (0). Also, movement end and 100% of MT are notionally the same but “movement end” was cut on line 326 (now line 334).
-L 445. You say that the “Rightward group deviated the most at movement end”. However, the amplitudes of these deviations were similar to that of the Lefward group at this time. Therefore, it is not clear that sensory processing was different in the two groups of subjects. L 446-442. These information are interesting but maybe beside the point, I suggest to focus the discussion on the results of the present study.
Please recall that we reported the absolute DB values. The signed average deviation for the rightward group at 100% of MT (or movement end) in the complete condition was of 7.7 mm vs. -4.6 mm for the leftward group. Be it more, the leftward group exhibited average -12.0 mm bias at 40% of movement time in the complete condition vs. 2.6 mm for the rightward group. Nevertheless, the question implies that it was important to remind the reader that the analyses were made on the absolute (thus unsigned) trajectory biases data (see lines 463-464).
The discussion is short compared to the introduction and should be improved.
The introduction has 1560 words and the discussion has 2410 words. As well, we have created a figure to help support the discussion and believe that we have addressed all of the critical points that needed to be discussed. Still, if there is a specific improvement that is required, we will be glad to make it.
Reviewer 2 Report
Dear Authors,
your manuscript reports an elegant experiment in the neurophysiology of movement.
However, I note that it is the repetition of what is contained in the doctoral thesis of one of you (Alekhina M., 2011: The role of neck muscles afferentation in planning and online control of goal-directed movement. Doctoral dissertation, University of Toronto). The comparison with the subsequent acquisitions relating to the topic dealt with is limited to 4 references subsequent to 2011. Figure 1, figure 5 and table 1 show only slight variations with respect to what is present in the thesis; while figures 3 and 4 refer to a smaller number of subjects than those described, in analogous figures, in the dissertation.
As far as I know, the thesis has not been published in a journal. But I consider it appropriate the - not present in the current version of the manuscript - citation of the original document.
Regarding the references:
Manson et al. (2019) - line 118 - is not included in the list of "References";
under item [42] two papers are erroneously reported: Smith et al. (2017) and McIntyre et al. (2007).
Best regards.
Author Response
We thank Reviewer 2 for their positive and constructive comments on our manuscript. There were many important elements to clarify and refine. Below, we present the original comments as well as our responses. As well, please note that we have made a few more edits, including the citation, correcting a few typos, and removing two references not cited in the text. The updated manuscript format has been edited with track-changes, to highlight all changes since the last submitted version.
Reviewer 2
Dear Authors,
your manuscript reports an elegant experiment in the neurophysiology of movement.
However, I note that it is the repetition of what is contained in the doctoral thesis of one of you (Alekhina M., 2011: The role of neck muscles afferentation in planning and online control of goal-directed movement. Doctoral dissertation, University of Toronto). The comparison with the subsequent acquisitions relating to the topic dealt with is limited to 4 references subsequent to 2011. Figure 1, figure 5 and table 1 show only slight variations with respect to what is present in the thesis; while figures 3 and 4 refer to a smaller number of subjects than those described, in analogous figures, in the dissertation.
As far as I know, the thesis has not been published in a journal. But I consider it appropriate the - not present in the current version of the manuscript - citation of the original document.
Excellent point. We have now clarified that this work was part of Maria Alkhina’s thesis (see lines 610-611) and added the associated reference (see lines 616-617).
Regarding the references:
Manson et al. (2019) - line 118 - is not included in the list of "References";
Thank you very much for noting our omission. The reference has now been added (see lines 670-672).
under item [42] two papers are erroneously reported: Smith et al. (2017) and McIntyre et al. (2007).
Indeed. Thanks for catching this inappropriate insertion of the McIntyre reference, which has been moved in its appropriate place (see line 680).
Best regards.
Reviewer 3 Report
I suggest that the authors use a better topic for the paper.
the references in the text are not based on the journal’s format.
participants should be clarified more in detail. Inclusion and exclusion criteria, formular for sample size calculation, etc….
Please add some practical implications for the findings of this paper.
Author Response
We thank Reviewer 3 for their positive and constructive comments on our manuscript. There were many important elements to clarify and refine. Below, we present the original comments as well as our responses. As well, please note that we have made a few more edits, including the citation, correcting a few typos, and removing two references not cited in the text. The updated manuscript format has been edited with track-changes, to highlight all changes since the last submitted version.
Reviewer 3
I suggest that the authors use a better topic for the paper.
We did consider many different ways to share the findings of this study and feel that the current approach is most appropriate. Notably, our goal was to investigate the impact of neck vibration on limb trajectories (i.e., prior to, during, and both), which seems to have been reached. Still, we do welcome specific suggestions on how to further improve the manuscript or specific criticisms as to why the current state is problematic. And if the reviewer was referring to the title, please let us know and we would be OK to change it.
the references in the text are not based on the journal’s format.
Agreed. The format was inappropriate at the time of submission but we believe that it has now been appropriately corrected (see updated document, with new format).
participants should be clarified more in detail. Inclusion and exclusion criteria, formular for sample size calculation, etc….
Thank you for asking. The inclusion criteria were present but we have now added the exclusion criteria (see lines 154-155). In terms of the sample size calculation, we failed to perform any at the outset of the study. However, as per the post-hoc power calculation mentioned for the first major comment of Reviewer 1, it appears that the study was appropriately powered in terms of the total sample size.
Please add some practical implications for the findings of this paper.
We made a modest and humble attempt to do so on lines 594-596.
Round 2
Reviewer 1 Report
Thank you for your responses.
Probably that the uploaded manuscript (brainsci-2577867-peer-review-v2.pdf) is not the good one:
-You said that “the updated manuscript format has been edited with track-changes, to highlight all changes since the last submitted version”, but there are no highlights.
-You answered that “We clarified in the figure caption that the traces were for the average of the trials (see line 296). And we hope that increasing the size of Figure 4 helped identify the error bars.” But I can see no changes.
I see no addition to lines 194 or 226-227…
However, the authors answered my major comments. My confusion about the timing of the vibration came from your figure 2 where the times corresponding to the movement start and movement end are precisely indicated.
I have no more comments.
Congratulations for your work.
Author Response
Thank you for flagging that I failed to upload the proper file (and my apologies for that error). The new manuscript should be updated now (filename: brainsci-2577867-author_lt_gm_jb_v2.docx).
The attached document should have all previously specified changes. As well, arising from this latest comment about Figure 2, the caption has been amended to clarify that the values at the top are averages.
Many thanks for the thorough and constructive review!
